# Weight and body composition outcomes with liraglutide in individuals with well-treated hypothyroidism: A retrospective case-control study

Tariq Chukir[1,2*], Mohammad Yaghmour[1], Turki Almutairi[3], Odette Chagoury[2], Shahrad Taheri[1,2,4]

1 Medical Education, Weill Cornell Medicine – Qatar, Doha, Qatar, 2 National Obesity Treatment Center, Hamad Medical Corporation, Doha, Qatar, 3 Department of Anesthesiology, Case Western Reserve University/University Hospitals – Cleveland Medical Center, Cleveland, Ohio, United States of America, 4 Diabetes Research Centre, University of Leicester, Leicester, United Kingdom

* tac2007@qatar-med.cornell.edu

## Abstract

### Background

Hypothyroidism is associated with weight gain. Although levothyroxine therapy restores thyroid hormone levels, its impact on weight loss is modest. Whether individuals with appropriately treated hypothyroidism respond differently to obesity pharmacotherapy remains relative unexplored. This study aimed to evaluate weight and body composition responses to liraglutide in patients with obesity and levothyroxine-treated hypothyroidism compared to matched controls without thyroid disease.

### Methods

In this retrospective case-control study, we included adults with overweight or obesity who were treated with liraglutide between 2017 and 2022 at an academic weight management center. Participants with hypothyroidism on stable levothyroxine therapy were compared to age-, gender-, and BMI-matched controls without thyroid disease. Outcomes include percentage weight change, categorical weight loss thresholds and changes in body composition assessed via bioelectrical impedance. Univariate and multivariate regression were used to identify predictors of weight loss.

### Results

Fifty-three patients with hypothyroidism were compared to 145 controls. After a mean follow-up of 10 months, the percent weight change was similar between groups (−10.8% vs. −8.9%, P = 0.940), as was the proportion of weight reduction from fat-free mass loss (29.9% vs. 33.3%, P = 0.729). A significant proportion in both groups achieved ≥5% weight loss (79.2% vs. 71.0%, P = 0.248). Baseline thyroid stimulating

**Data availability statement:** The data that support the findings of this study are not publicly available due to data that could compromise the privacy of research participants. Restrictions on public sharing of data have been imposed by Hamad Medical Corporation, Qatar to protect confidentiality. Data access requests can be submitted to the institution and are subject to institutional approvals. Institutional email address: qmi@hamad.qa.

**Competing interests:** The authors have declared that no competing interests exist.

hormone (TSH) levels within the normal range did not correlate with weight outcomes. In univariate analysis, baseline metformin use was associated with greater weight loss (P = 0.042), but this was not significant in multivariate models.

## Conclusion

Liraglutide leads to clinically meaningful and comparable weight loss in individuals with well-treated hypothyroidism and those without thyroid disease. Body composition changes were also similar between groups. These findings help clinicians address a frequently asked question among patients with hypothyroidism about their ability to effectively lose weight.

## Introduction

Obesity is a complex and multifactorial disease that is associated with numerous health-related complications. Frequently, obesity either causes or exacerbates other medical conditions. Similarly, several medical conditions and/or their treatments may result in weight gain. Responses to treatment for obesity is influenced by other accompanying medical conditions. For example, type 2 diabetes mellitus is associated with a lower weight response to obesity treatments. Thyroid underactivity is recognised as a secondary cause of obesity and as a factor that might reduce the degree of weight reduction response to obesity treatments [1–3]. Weight gain with hypothyroidism is multifactorial and has been proposed to be related to a reduction in energy expenditure, alterations in substrate metabolism, and body fluid changes [2,3]. Hypothyroidism is treated with levothyroxine therapy but weight reduction after hormone replacement appears to be modest and mainly associated with fluid reduction and reduction in lean mass rather than fat mass [4]. Obesity may impair adequate thyroid hormone replacement and weight reduction may alter levothyroxine dosage requirements. For example, levothyroxine dose requirement may decrease after bariatric surgery due reduced adipose tissue distribution of levothyroxine [5]. The levothyroxine requirement, however, may increase in some individuals after bariatric surgery due to reduced intestinal absorption [6].

Frequently, individuals with biochemically adequate levothyroxine replacement express struggles with achieving adequate weight reduction with obesity interventions. However, the weight loss response to glucagon-like peptidde- 1 (GLP-1) receptor agonists such as liraglutide in patients with treated hypothyroidism remains relatively unexplored. To investigate this, we conducted a retrospective case-control study to examine weight and body composition responses of people with obesity and hypothyroidism treated with levothyroxine.

## Methods

This study has been granted an exemption from full review by the Institutional Review Boards at Weill Cornell Medicine – Qatar (IRB 22–00015) and Hamad Medical Corportation (MRC-01-22-550). Consent was waived for this retrospective study.

This retrospective case-control study investigated body weight and composition changes in people aged 18 years and older living with overweight or obesity and a history of levothyroxine-treated hypothyroidism who were initiated on the glucagon-like peptide-1 (GLP-1) receptor agonist liraglutide 3.0 mg for weight management at an academic weight management center between 2017 and 2022 and had at least 1 follow-up visit. Data were reviewed and extracted from the electronic medical records until Oct. 1st 2024. Liraglutide dose titration varied based on patient's response and tolerability. After data was collected, data was de-identified during analysis and Interpretation. A control group of age-, gender-, body mass index (BMI)-matched individuals were identified. Exclusions included: no follow up after 6–12 months, bariatric surgery during the follow up period, abnormal thyroid stimulating hormone (TSH) levels during the follow up period, or alternative treatments, e.g., semaglutide or tirzepatide. Clinical data, including demographics, comorbidities, bodyweight, body composition, biochemical laboratory data and medications were extracted from the electronic medical records. Body composition was assessed by bioelectrical impedance analysis using Tanita (Dual-frequency Body Composition Analyser – DC360, Tanita, Japan). All those included received counselling on lifestyle modifications provided by physicians and/or dietitians.

The primary outcome was the percentage change in bodyweight between the initial visit and the follow-up closest to one year, as well as categorical weight reduction, defined as the proportion achieving ≥5%, ≥ 10%, or ≥15% weight reduction. The secondary outcomes included the proportion of weight reduction attributed to fat-free mass (FFM) and the correlation between the percentage change in weight and certain clinical variables, including TSH level, bariatric surgery, history of type 2 diabetes and the use of metformin.

Descriptive statistical analyses of baseline characteristics and outcomes were performed. Comparisons between variables were performed using t-tests, Mann-Whitney U tests, and chi-square tests, as appropriate. Univariate and multivariate linear regression was used to assess for correlation between independent variables and the final percentage change in weight. Since levothyroxine dose and TSH could be correlated, including both could cause multicollinearity in multivariate regression. We selected TSH to be included in the multivariate regression because it reflects the degree of thyroid hormone replacement achieved with levothyroxine. A P value of <0.05 was considered as statistically significant. All statistical analyses were conducted using R Studio (Version 2024.09.1+394).

## Results

A total of 2595 patients with overweight or obesity were on initiated on liraglutide between 2017 and 2022 and had at least 1 follow-up visit, of whom 53 were treated with levothyroxine and met the inclusion criteria. This group was compared to 145 age-, gender-, BMI- matched controls. The 2 groups had similar mean age [38.8±9.4 vs.38.7±9.2 (P=0.907)], mean BMI [37.0±4.6 vs. 35.7±4.9 (P=0.088)] and mean TSH levels [2.5±2.1 vs. 2.1±1.1(P=0.168)]. There was no significant difference in any of the other baseline characteristics between both groups (Table 1).

After a mean follow-up of 10 months, the mean final percent weight change was −10.8±6.5 and −8.9±7.9 (P=0.940) in the levothyroxine-treated and the control group, respectively. In addition, the mean final proportion of weight reduction from FFM was 29.9±23.5% vs. 33.3±25.2% (P=0.729) (Table 2).

The majority of participants achieved a weight reduction of >5% [42 (79.2%) vs. 103 (71.0%),P=0.248 (Fig 1).

In a univariate linear regression model, metformin use correlated with the final percentage weight reduction (estimate=−3.7, SE=2.9, 95% CI [−7.3 to −0.1], P=0.042. Other variables, e.g., TSH levels and levothyroxine dose did not significantly correlate. In a multivariate linear regression model, baseline TSH (estimate=0.5, SE=0.4, 95%CI [−0.3–1.3], P=0.216, type 2 diabetes (estimate=−5.4, SE=3.8, 95% CI[−13.0, 2.2], P=0.161), history of bariatric surgery (estimate=−3.4, SE=2.0, 95% CI [−7.5, 0.6],P=0.097), and the use of metformin (estimate=−3.2, SE=1.8, 95% CI[−6.8, 0.4],P=0.079) were not significantly correlated with the final percentage weight change (Table 3).

## Discussion

In this study, we demonstrated that liraglutide-induced weight reduction in individuals with adequately treated hypothyroidism was clinically significant and comparable to outcomes observed in age-, gender-, and BMI-matched controls without

**Table 1. Baseline Characteristics of individuals with hypothyroidism (n = 53) and individuals with normal thyroid function (n = 145).**

|  | Hypothyroidism n = 53 | Normal thyroid function n = 145 | P-value[a] |
|---|---|---|---|
| Age in Years, mean ± SD | 38.8 ± 9.4 | 38.7 ± 9.2 | 0.907 |
| Female sex, n (%) | 48 (90.6) | 133 (91.7) | 0.999 |
| Baseline Weight in Kg, mean ± SD | 95.4 ± 15.2 | 91.3 ± 14.0 | 0.096 |
| Baseline Body Mass Index in Kg/m², mean ± SD | 37.0 ± 4.6 | 35.7 ± 4.9 | 0.088 |
| Body Mass Index Category |  |  |  |
| Overweight | 1 (1.9) | 8 (5.5) |  |
| Class 1 Obesity | 17(32.1) | 64 (44.1) |  |
| Class 2 Obesity | 24 (45.3) | 46 (31.7) |  |
| Class 3 Obesity | 11 (20.8) | 29 (20) |  |
| Baseline Fat Free Mass Percentage, mean ± SD | 55.9 ± 6.3 | 57.1 ± 5.5 | 0.286 |
| Baseline Thyroid Stimulating Hormone, median [q1, q3] | 2.5 ± 2.1 | 2.1 ± 1.1 | 0.168 |
| Type 2 diabetes, n (%) | 3 (5.7) | 2 (1.4) | 0.235 |
| Prediabetes, n (%) | 10 (18.9) | 19 (13.3) | 0.524 |
| Haemoglobin A1C, mean ± SD | 5.5 ± 0.4 | 5.5 ± 0.4 | 0.637 |
| Hypertension, n (%) | 4 (7.5) | 14 (9.7) | 0.849 |
| Metabolic dysfunction-associated steatotic liver disease, n (%) | 1 (1.9) | 1 (0.7) | 0.636 |
| History of bariatric surgery, n (%) | 12 (22.6) | 51 (35.2) | 0.133 |
| Metformin use, n (%) | 33 (62.3) | 84 (58.3) | 0.738 |
| SGLT2i use, n (%) | 2 (3.8%) | 0 | 0.122 |
| Following with dietitian, n (%) | 16 (30.2) | 33 (25.6) | 0.008 |

Continuous variables are presented as mean ± standard deviation (SD), except for thyroid-stimulating hormone (TSH), which is reported as median [Q1, Q3]. Categorical variables are presented as number (percentage).

Comparisons between continuous variables were performed using independent t-tests or Mann-Whitney U tests, while categorical variables were analyzed using chi-square tests.

Abbreviation: SGLT2 = sodium-glucose transport protein 2.

**Table 2. Weight and body composition outcomes between individuals with hypothyroidism (n = 53) and individuals with normal thyroid function (n = 145).**

|  | Hypothyroidism n = 53 | Normal thyroid function n = 145 | P-value |
|---|---|---|---|
| Final percentage weight change, mean ± SD | −10.8 ± 6.5 | −8.9 ± 7.9 | 0.940 |
| Duration of follow up in days, mean ± SD | 303 ± 113.4 | 295.4 ± 115.5 | 0.338 |
| Final Proportion of Weight Reduction From Fat Free Mass in percent, mean ± SD | 29.9 ± 23.5 | 33.3 ± 25.2 | 0.729 |

Continuous variables are presented as mean ± standard deviation (SD). Groups were compared using t-tests.

thyroid disease. These findings are consistent with data from bariatric surgery studies, which have demonstrated significant weight reduction in people with a history of hypothyroidism following bariatric surgery [7].

Additionally, we found that the proportion of weight reduction attributable to fat-free mass was similar between the two groups and consistent with magnitude reported in the literature with other weight reduction interventions (approximately 20–30% of the weight loss is from lean mass) [8]. A previous study demonstrated that the weight reduction in response to initiating levothyroxine in patients with newly diagnosed primary hypothyroidism primarily occurs due to the reduction of excess body water, with minimal fat reduction [4]. Our findings suggest that individuals with appropriately treated hyothyroidism on maintenance therapy exhibit similar changes in body composition in response to weight reduction to those without thyroid disease.

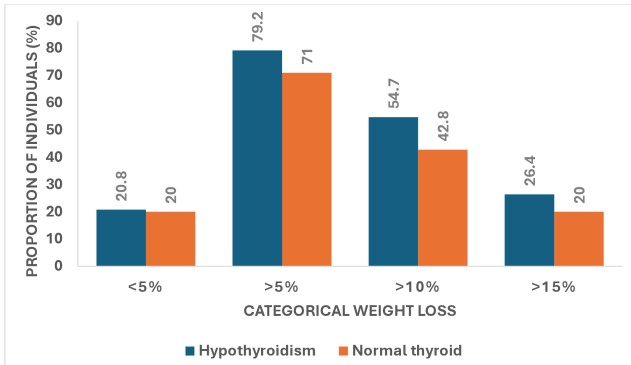

**Fig 1. Categorical weight loss in individuals with levothyroxine-treated hypothyroidism compared to individuals with normal thyroid function.** Outcomes were similar in all categories. Chi-square test was used to compare groups. A P value of <0.05 was considered as statistically significant.

**Table 3. Linear Regression Analysis assessing for correlation between selected independent variables and final percentage weight change.**

| Variable | Univariate | | | | Multivariate± | | | |
|---|---|---|---|---|---|---|---|---|
| | Estimate | SE* | 95% CI£ | P-value | Estimate | SE | 95%CI | P-value |
| TSH | 0.5 | 0.4 | −0.3 to 1.3 | 0.255 | −0.5 | 0.4 | −0.3 to 1.3 | 0.216 |
| T2DM | −6.3 | 3.8 | −13.9 to 1.4 | 0.106 | −5.4 | 3.8 | −13.0 to 2.2 | 0.161 |
| Bariatric surgery | −3.1 | 2.1 | −7.3 to 1.2 | 0.151 | −3.4 | 2.0 | −7.5 to 0.6 | 0.097 |
| Metformin | −3.7 | 1.8 | −7.3 to −0.1 | 0.042 | −3.2 | 1.8 | −6.8 to 0.4 | 0.079 |

Values are presented as estimated regression coefficients with standard error (SE), 95% confidence intervals (CI), and P values. P-values from the linear regression model reflect the statistical significance of each variable's association with the outcome variable (final percentage weight change).

* standard error.

£ confidence interval.

± Multivariate linear regression model included all 4 variables.

Abbreviations: TSH= thyroid stimulating hormone, T2DM = type 2 diabetes.

Furthermore, there is currently no strong evidence to suggest that lower TSH levels within the normal range correlate with greater weight reduction. Previous studies have reported no association between baseline TSH levels and weight reduction outcomes [9]. Consistent with these findings, our study did not identify a correlation between TSH levels and weight reduction outcomes in patients treated with levothyroxine. Our study suggested that being on metformin at baseline could predict greater weight reduction, a finding that was observed with tirzepatide in a posthoc analysis of the SURPASS trial [10]. It is important to note that the analysis may have been underpowered by the study's limited sample size, potentially reducing the ability to detect statistical significance in the multivariate linear regression analysis.

Our study has several strengths, including the use of data from a real-world setting, which enhances the generalizability of the findings to clinical practice. Additionally, the assessment of body composition provides valuable insights beyond traditional weight reduction metrics. However, several limitations must be acknowledged due to the retrospective nature of the study, including the potential for selection bias and confounding. For instance, factors such as dietary adherence and physical activity levels may have influenced the outcomes but could not be reliably assessed because data on these variables were not available. Furthermore, data on socioeconomic status, a well-recognized determinant of health outcomes in many populations, was not available. Of note, Qatar has a universal healthcare coverage that provides free access to obesity managment medications for people living with obesity. All patients in this study were prescribed the 3.0 mg dose of

liraglutide and more than 90% of participants in both groups were able to reach the maximal dose. While some dose titration based on clinical judgment may have occurred, it was not consistently documented in the medical records. However, any such variation is likely to have occurred at a similar rate across both groups.

The findings of this research have clinical significance, offering valuable insights for patients with hypothyroidism regarding expected responses to liraglutide when used as an adjunct to lifestyle modification for weight management. Those achieving greater weight reduction may require adjustments to their levothyroxine dose. Currently, there is no established guidelines on medication adjustments in patients undergoing weight reduction and should be the focus of future research, particularly with newer generation anti-obesity medications.

## Conclusion

In conclusion, our study demonstrates that individuals with well-treated primary hypothyroidism experience clinically significant and comparable weight reduction with liraglutide to those without thyroid disease. Importantly, changes in body composition were similar between groups. These findings support the effectiveness of liraglutide in this population. Our results may help clinicians address a common concern among patients with hypothyroidism regarding their ability to lose weight effectively.

## Acknowledgments

This publication was made possible by the support provided from Hamad Medical Corporation and Weill Cornell Medicine – Qatar. The findings herein reflect the work and are solely the responsibility of the authors.

## Author contributions

**Conceptualization:** Tariq Chukir, Shahrad Taheri.

**Data curation:** Tariq Chukir, Mohammad Yaghmour, Turki Almutairi.

**Formal analysis:** Tariq Chukir, Mohammad Yaghmour.

**Investigation:** Tariq Chukir.

**Methodology:** Tariq Chukir.

**Project administration:** Tariq Chukir, Odette Chagoury.

**Supervision:** Tariq Chukir, Shahrad Taheri.

**Writing – original draft:** Tariq Chukir.

**Writing – review & editing:** Tariq Chukir, Mohammad Yaghmour, Odette Chagoury, Shahrad Taheri.

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
