## [Decision Letter · Decision Letter 0]

23 Jul 2025

Dear Dr. Chukir,

Thank you for submitting your manuscript to PLOS ONE. After careful consideration, we feel that it has merit but does not fully meet PLOS ONE’s publication criteria as it currently stands. Therefore, we invite you to submit a revised version of the manuscript that addresses the points raised during the review process.

We look forward to receiving your revised manuscript.

Kind regards,

Afagh Hassanzadeh Rad

Academic Editor

PLOS ONE

Journal Requirements:

3. In the online submission form you indicate that your data is not available for proprietary reasons and have provided a contact point for accessing this data. Please note that your current contact point is a co-author on this manuscript. According to our Data Policy, the contact point must not be an author on the manuscript and must be an institutional contact, ideally not an individual. Please revise your data statement to a non-author institutional point of contact, such as a data access or ethics committee, and send this to us via return email. Please also include contact information for the third party organization, and please include the full citation of where the data can be found.

“The publication of this article was funded by the Qatar National Library”

5. Thank you for stating the following in the Competing Interests/Financial Disclosure section:

We note that one or more of the authors are employed by a commercial company: Hamad Medical Corporation

Reviewers' comments:

Reviewer's Responses to Questions

**Comments to the Author**

1. Is the manuscript technically sound, and do the data support the conclusions?

Reviewer #1: Yes

Reviewer #2: Yes

2. Has the statistical analysis been performed appropriately and rigorously?

Reviewer #1: Yes

Reviewer #2: Yes

3. Have the authors made all data underlying the findings in their manuscript fully available?

Reviewer #1: Yes

Reviewer #2: Yes

4. Is the manuscript presented in an intelligible fashion and written in standard English?

Reviewer #1: Yes

Reviewer #2: Yes

Reviewer #1: Dear Editor and Authors,

First, thank you for the opportunity to review this manuscript.

I have carefully reviewed the manuscript, which presents a clearly written and clinically relevant retrospective case-control study examining the effects of liraglutide on weight and body composition in patients with well-managed hypothyroidism. The article is well-structured and conforms to the standards expected for research publications in PLOS ONE. The abstract effectively summarizes the research question, methodology, results, and conclusions. The clinical research question is clearly articulated and appropriately justified.

However, several important issues should be addressed to improve the quality and clarity of the manuscript:

Please clarify whether the dose of liraglutide was consistent across all participants or individually adjusted based on clinical conditions. This detail is essential for an accurate interpretation of the weight loss outcomes.

While a multivariate regression analysis was conducted, certain lifestyle-related variables—such as dietary adherence, physical activity levels, or treatment compliance—may have a significant impact on the results. If such data are available, please include them; otherwise, their absence should be acknowledged in the limitations section.

Some sentences in the Discussion section would benefit from linguistic editing to enhance clarity and fluency.

The title of Figure 1 should include a brief description of the statistical test used.

Please resolve the discrepancies between the list of figures and tables and their corresponding captions within the manuscript.

Ensure consistent formatting of p-values throughout the manuscript.

Thank you again for the opportunity to review this work.

Sincerely

dr. matin mojaveri samak

Reviewer #2: dear authors

your manuscript was interesting.

it is recommended to include the levothyroxine dose in multivariate regression model as well. for example up to 50,up to 100 , up to 150...

all the best

**Do you want your identity to be public for this peer review?** For information about this choice, including consent withdrawal, please see our Privacy Policy

Reviewer #1: **Yes: ** matin mojaveri samak

Reviewer #2: **Yes: ** zahra abbasi ranjbar

---

## [Author Response · Author response to Decision Letter 1]

29 Jul 2025

Dear Drs. Afagh Hassanzadeh Rad, Matin Mojaveri Samak and Zahra Abbasi Ranjbar,

Thank you for your thoughtful comments. We believe your comments and suggestions strengthen our manuscript and provide clinically relevant insights.

Reviewer 1: Dr. Matin Mojaveri Samak

Comment 1: Please clarify whether the dose of liraglutide was consistent across all participants or individually adjusted based on clinical conditions. This detail is essential for an accurate interpretation of the weight loss outcomes.

Response 1: Thank you for your insightful comment. We agree that clarifying the liraglutide dosing is important for accurately interpreting the weight loss outcomes. We have now added this information to the Methods and Discussion sections of the manuscript.

The following sentence has been added to the manuscript: "All patients were prescribed the 3.0 mg dose of liraglutide and more than 90% of participants in both groups were able to reach the maximal dose. While some dose titration based on clinical judgment may have occurred, it was not consistently documented in the medical records. However, any such variation is likely to have occurred at a similar rate across both groups."

Comment 2: While a multivariate regression analysis was conducted, certain lifestyle-related variables—such as dietary adherence, physical activity levels, or treatment compliance—may have a significant impact on the results. If such data are available, please include them; otherwise, their absence should be acknowledged in the limitations section.

Response 2: We agree that other variables, such as dietary adherence and physical activity levels, may have an impact on the final outcomes. We have added this limitation to the discussion section. It is worth noting that even many phase 3 randomized controlled trials of obesity pharmacotherapy, unfortunately, often do not report adherence to lifestyle recommendations. The following sentence has been added to the manuscript:

“several limitations must be acknowledged due to the retrospective nature of the study, including the potential for selection bias and confounding. For instance, factors such as dietary adherence and physical activity levels may have influenced the outcomes but could not be reliably assessed because data on these variables were not available.”

Comment 3: Some sentences in the Discussion section would benefit from linguistic editing to enhance clarity and fluency.

Response 3: Thank you for pointing this out. The discussion was reviewed and edited to ensure clarity and fluency.

Comment 4: The title of Figure 1 should include a brief description of the statistical test used.

Response 4: A brief description has been added.

“Fig 1. Categorical weight loss in individuals with levothyroxine-treated hypothyroidism compared to individuals with normal thyroid function. Outcomes were similar in all categories. Chi-square test was used to compare groups. A p value of <0.05 was considered as statistically significant.”

Comment 5: Please resolve the discrepancies between the list of figures and tables and their corresponding captions within the manuscript.

Response 5: Thank you for pointing this out. Reviewed and resolved.

Comment 6: Ensure consistent formatting of p-values throughout the manuscript.

Response 6: Reviewed and resolved

Reviewer 2:

Comment 1: It is recommended to include the levothyroxine dose in multivariate regression model as well. for example up to 50,up to 100 , up to 150...

Response 1: Thank you for your suggestion.

We ran the model using levothyroxine dose as a linear variable as well as a categorical variable but we did not find any significance in our data. Since levothyroxine dose and TSH could be correlated, including both could cause multicollinearity in multivariate regression. We selected TSH to be included in the multivariate regression because it reflects the degree of thyroid hormone replacement achieved with levothyroxine. We elaborated on this in the manuscript in methods and results.

---

## [Decision Letter · Decision Letter 1]

20 Aug 2025

Dear Dr. Chukir,

Thank you for submitting your manuscript to PLOS ONE. After careful consideration, we feel that it has merit but does not fully meet PLOS ONE’s publication criteria as it currently stands. Therefore, we invite you to submit a revised version of the manuscript that addresses the points raised during the review process.

We look forward to receiving your revised manuscript.

Kind regards,

Afagh Hassanzadeh Rad

Academic Editor

PLOS ONE

Journal Requirements:

Reviewers' comments:

Reviewer's Responses to Questions

**Comments to the Author**

Reviewer #1: All comments have been addressed

Reviewer #3: All comments have been addressed

2. Is the manuscript technically sound, and do the data support the conclusions?

Reviewer #1: Yes

Reviewer #3: Yes

3. Has the statistical analysis been performed appropriately and rigorously?

Reviewer #1: Yes

Reviewer #3: Yes

4. Have the authors made all data underlying the findings in their manuscript fully available?

Reviewer #1: Yes

Reviewer #3: Yes

5. Is the manuscript presented in an intelligible fashion and written in standard English?

Reviewer #1: Yes

Reviewer #3: Yes

Reviewer #1: (No Response)

Reviewer #3: Thank you for submitting your revised manuscript. I appreciate your careful attention to the reviewers' comments and the improvements made to the manuscript, which is now much stronger and addresses the key concerns raised. However, a few minor clarifications and editorial adjustments are still needed before final acceptance:

1.Please explicitly state in the Methods whether dose titration followed a standardized protocol or was clinician dependent.

2.Expand the discussion of limitations to include potential unmeasured confounders (e.g., socioeconomic factors, adherence to lifestyle interventions).

3.Ensure uniform formatting of p-values (e.g., "p = 0.04" , "p<0.05") and clarify the discrepancy between univariate and multivariate results for metformin. Moreover, Each table should speak for itself, so please include complete notes for each table (e.g., abbreviations and how p-values were determined) in all of them.

Once addressed, the manuscript will be suitable for acceptance.

We look forward to receiving your final version.

**Do you want your identity to be public for this peer review?** For information about this choice, including consent withdrawal, please see our Privacy Policy

Reviewer #1: **Yes: ** matin mojaveri samak

Reviewer #3: No

---

## [Author Response · Author response to Decision Letter 2]

21 Aug 2025

Dear Drs. Afagh Hassanzadeh Rad, and reviewers,

Thank you for your thoughtful comments. We believe your comments and suggestions strengthen our manuscript and provide more clinically relevant insights.

Reviewer #3: Thank you for submitting your revised manuscript. I appreciate your careful attention to the reviewers' comments and the improvements made to the manuscript, which is now much stronger and addresses the key concerns raised. However, a few minor clarifications and editorial adjustments are still needed before final acceptance:

1.Please explicitly state in the Methods whether dose titration followed a standardized protocol or was clinician dependent.

Thank you for this important comment comment. We have clarified this point in the Methods section. “. Liraglutide dose titration varied based on patient’s response and tolerability.”

2.Expand the discussion of limitations to include potential unmeasured confounders (e.g., socioeconomic factors, adherence to lifestyle interventions).

Thank you for pointing out these limitations. The discussion has been expanded.

“However, several limitations must be acknowledged due to the retrospective nature of the study, including the potential for selection bias and confounding. For instance, factors such as dietary adherence and physical activity levels may have influenced the outcomes but could not be reliably assessed because data on these variables were not available. Furthermore, data on socioeconomic status, a well-recognized determinant of health outcomes in many populations, was not available. Of note, Qatar has a universal healthcare coverage that provides free access to obesity managment medications for people living with obesity.”

3.Ensure uniform formatting of p-values (e.g., "p = 0.04" , "p<0.05") and clarify the discrepancy between univariate and multivariate results for metformin. Moreover, Each table should speak for itself, so please include complete notes for each table (e.g., abbreviations and how p-values were determined) in all of them.

Thank you for your feedback. Uniform formating of P-values is reported now.

We also modified the discussion to clarify our results.

“Our study suggested that being on metformin at baseline could predict greater weight reduction, a finding that was observed with tirzepatide in a posthoc analysis of the SURPASS trial [10]. It is important to note that the analysis may have been underpowered by the study’s limited sample size, potentially reducing the ability to detect statistical significance in the multivariate linear regression analysis.”

Notes were also added below all tables to improve readibility.

---

## [Editor Report · Decision Letter 2]

26 Aug 2025

Weight and body composition outcomes with liraglutide in individuals with well-treated hypothyroidism: a retrospective case-control study

PONE-D-25-21992R2

Dear Dr. Chukir,

We’re pleased to inform you that your manuscript has been judged scientifically suitable for publication and will be formally accepted for publication once it meets all outstanding technical requirements.

Kind regards,

Afagh Hassanzadeh Rad

Academic Editor

PLOS ONE
---

## [Editor Report · Acceptance letter]

PONE-D-25-21992R2

PLOS ONE

Dear Dr. Chukir,

I'm pleased to inform you that your manuscript has been deemed suitable for publication in PLOS ONE. Congratulations! Your manuscript is now being handed over to our production team.

Kind regards,

on behalf of

Dr. Afagh Hassanzadeh Rad

Academic Editor

PLOS ONE